# Prediction for the Influence of Guide Vane Opening on the Radial Clearance Sediment Erosion of Runner in a Francis Turbine

**Zhiqiang Jin [1], Xijie Song [2], Anfu Zhang [1], Feng Shao [1] and Zhengwei Wang [2,\*]**

1 Xinjiang Xinhua Hydroelectric Investment Technology Co., Ltd., Akesu 843300, China
2 State Key Laboratory of Hydroscience and Engineering, Department of Energy and Power Engineering, Tsinghua University, Beijing 100084, China
\* Correspondence: wzw@mail.tsinghua.edu.cn

**Abstract:** In this paper, the Eulerian–Lagrangian method and Tabakoff erosion model are used to study the solid–liquid two-phase flow in a Francis turbine. Through the analysis of the overall flow pattern, particle flow, particle concentration, and wear in the bladeless area of the unit under different guide vane openings, the influence of runner radial gap flow on the surrounding flow field characteristics and wear under different guide vane openings is revealed. The results show that the smaller the opening of the guide vane, the greater the influence on the vortices and flow pattern and the particle distribution in the runner. The overall wear in the hydraulic turbine unit with the optimal opening is the smallest. The long-term wear of the runner inlet and guide vane outlet will cause the loss of local structures, an increase in the radial clearance of the runner, an increase in the clearance leakage, an increase in the vibration of the unit, and a reduction in efficiency. The research results provide a basis for the structural and hydraulic optimization of the Francis turbine.

**Keywords:** Francis turbine; guide vane opening; sediment erosion; clearance; CFD

## 1. Introduction

China's rivers contain a lot of sand; moreover, 115 rivers have an average annual sediment discharge of more than 10 million tons. Regarding the Yangtze River, the average annual sediment discharge has reached 514 million tons; the average annual sediment concentration in the Three Gorges Reservoir Area is 1.17 kg/m$^3$ and the maximum is 10.5 kg/m$^3$. The sediment concentration in the Yellow River is higher. According to statistics, the average annual sediment concentration of the Sanmenxia reach of the Yellow River is 37.6 kg/m$^3$. The high sediment concentration in rivers in China leads to sediment abrasion of 30~40% of hydropower stations [1,2].

Hydraulic machinery has therefore been operating in the water flow with high sediment volume for a long time, which has caused different degrees of abrasion of the turbines of most power stations in China [2]. Severe wear causes damage to the structural materials of the hydropower equipment itself, affects the reliability and stability of the operation, reduces the efficiency and output of the turbine, shortens the service life of the runner, prolongs the construction period, increases the consumption of materials and spare parts, and causes huge economic losses [3,4].

During the rotation process of the Francis turbine, the pressure fluctuation in the non-blade area will be caused by the comprehensive influence of the flow field distortion caused by the runner blades and guide vanes [5]. The radial clearance of the runner is the only channel connecting the vaneless area with the lower cavity of the top cover and the upper cavity of the bottom ring, so the hydraulic characteristics in the vaneless area have a certain correlation with the hydraulic characteristics of the lower cavity of the top cover and the upper cavity of the bottom ring [6]. The wear in the clearance will change the clearance size, and the change in the clearance size will have an impact on the flow field characteristics around the runner [7]. The change in flow field characteristics around the

runner may lead to the increase in unit vibration and power swing, and even threaten the safe and stable operation of the power station [8].

Wear is the process of material transfer and loss in the contact surface layer during the relative movement of interacting solid surfaces. The working head of the hydraulic turbine is high, and the internal wear is mainly erosion wear [9]. In order to reveal the erosion wear mechanism, many researchers have studied the erosion wear from experimental and theoretical aspects, and put forward a variety of erosion wear prediction models. Finnie proposed the first erosion model. The model assumes that particles do not break up in the process of cutting metal, and considers that the erosion wear of solid particles on ductile materials is mainly due to the cutting effect of particles on materials [10]. Tabakoff introduces the impact angle and impact velocity to the base Finnie model, which has good prediction ability for the erosion and wear in the hydraulic turbine and has been widely used.

Sediment erosion mainly belongs to sediment-laden flow, which is a typical solid–liquid two-phase flow [11]. At present, a lot of studies have been performed on the Francis turbine, from the aspects of structure design, design method, basic equation, flow law, and wear mechanism. Li [12] studied the flow field characteristics of turbine guide vanes under the condition of sediment-laden flow, and found the best relative placement position of guide vanes and fixed guide vanes under the condition of sediment-laden flow. Liu [13] established the Euler–Lagrangian mixed turbulence model of low concentration solid–liquid two-phase flow, and produced the particle wall collision model and the erosion model of hydraulic turbine flow passage parts made of ductile metal materials. These models can be used to numerically simulate the flow of water containing sand in the hydraulic turbine, the concentration distribution of sand particles in the flow passage of the hydraulic turbine, the movement track, and the erosion rate of hydraulic turbine flow passage parts. Qi [14] derived the energy equation and energy dissipation term expression of solid–liquid two-phase flow describing sediment-laden flow from the energy equation of continuous medium flow, and introduced the energy dissipation extreme value principle into the study of sediment-laden flow characteristics in hydraulic turbines. Li [15] reported that water turbines in sediment-laden flow are prone to abrasion and damage. For the built hydropower station, adopting a reasonable operation mode and certain maintenance measures can delay or reduce the abrasion of the water turbine. Huang [16] expounded the micro process and mechanism of abrasion damage. The stability is related to the safe and normal operation of the unit. The sediment-laden flow is different from the single-phase flow of clean water. It is also necessary to explore the stability of the unit in sediment-laden flow.

The research background of this paper is Xinjiang Tagake Hydropower Station, which is located 14 km from the head of the Xiehera diversion canal in the Aksu region, Xinjiang. It is a runoff diversion hydropower station, with a diversion canal of 6.88 km long and a tailrace of 4.22 km long. Two mixed flow turbine generator units with a unit capacity of 24.5 MW are installed. The design diversion flow is 75 $m^3$/s, the rated head is 74 m, the guaranteed output is 13.8 MW, and the annual design power generation is 273.9 million kW·h. The high sediment concentration in Xinjiang makes the unit have serious sediment abrasion problem.

Due to the lack of research on clearance wear of hydraulic turbine, in this paper, numerical simulation is used to predict the sediment-laden flow and wear of a hydraulic turbine unit in the power station. A full fluid domain model of real machine size is established and the influence of runner radial clearance and clearance wear on the flow field characteristics around the runner under different guide vane openings was simulated through CFX, so as to improve the mechanical and hydraulic performance of the unit.

## 2. Mathematical and Geometrical Models

### 2.1. Mathematical Model

#### 2.1.1. Governing Equations

In this paper, the sediment laden flow is simulated and solved by Euler–Lagrange method. In the Euler–Lagrange system, the fluid phase is treated as a continuous phase,

solved by Euler methods, and the particle phase as a discrete phase, solved by Lagrange methods [17]. The flow control equation of continuous phase is solved by the *N-S* equation.

$$\frac{\partial(\rho u)}{\partial t} + \nabla \bullet (\rho u u) = -\nabla p + \rho v \Delta u - \rho \nabla \bullet \tau + S_t \tag{1}$$

where $u$ is the flow velocity, $t$ is the time, $\rho$ is the fluid density, $p$ is the flow pressure, $v$ is the kinematic viscosity of the fluid, and $S_t$ is the source term. $\tau$ is the Reynolds stress defined as:

$$\tau = \tau^d + \frac{2k}{3}\delta \tag{2}$$

where $\tau^d$ is the deviatoric Reynolds stress, k is the turbulent kinetic energy, $\delta$ is the Kronecker delta. Based on viscosity ($v_t$) assumptions, Equation (2) can be written as:

$$\tau = -2v_t S + \frac{2k}{3}\delta \tag{3}$$

where $S$ is the train-rate tensor,

$$S = \frac{1}{2}\left(\nabla u + \nabla^T u\right) \tag{4}$$

### 2.1.2. Lagrangian Tracking of Particle Motion

The particle movement in the hydraulic turbine is mainly represented as discrete phase movement, so the Lagrangian particle tracking model, which is widely used in sediment movement, is used to track the particle movement in the hydraulic turbine.

$$m_p \frac{du_p}{dt} = F_D + F_B + F_G + F_V + F_P + F_X \tag{5}$$

where $t$ is time, $m_p$ is particle mass, $u_p$ is particle velocity, $F_D$ is resistance, $F_B$ is Basset force, $F_G$ is gravity, $F_V$ is virtual mass force, $F_P$ is pressure gradient force, and $F_X$ is the sum of other external forces considered.

In this paper, the particle concentration in the flow field is small, the fluid velocity of the continuous phase in the pump is large, and there is a large density difference between the continuous phase and the discrete phase. Therefore, the virtual mass force, pressure gradient force, Basset force, Saffman force, and Magnus force on the solid particles can be ignored. The basic equation of particle motion can be expressed as:

$$\frac{dx_{\text{pi}}}{dt} = u_{\text{pi}} \tag{6}$$

$$\frac{du_{\text{pi}}}{dt} = \frac{3C_{D}\rho f}{4\rho_p D_p}|u_s|u_s \tag{7}$$

where $u_s$ is the slip velocity between particles and the liquid, $C_D$ is the drag coefficient related to Reynolds number, $\rho_f$ is the liquid density, $\rho_p$ is the particle density, $D_p$ is the particle diameter, and $x_{\text{pi}}$ is the spatial coordinate position of particles.

In this paper, one-way coupling between the fluid phase and the solid phase is adopted in the particle track model considering the particle concentration.

### 2.1.3. Erosion Model

The internal sediment erosion of hydraulic turbine is mainly impact wear, and the commonly used impact erosion models include the Finnie erosion model, Tabakoff erosion model, and Oka Erosion model [18]. Among them, the Tabakoff erosion model is widely used in the prediction of mixed flow turbine sediment erosion in engineering [19,20].

Therefore, in this paper, the Tabakoff erosion model is used to predict the sediment erosion in the turbine.

$$E = k_1 f(\gamma) V_P^2 \cos^2 \gamma \left[1 - R_T^2\right] + f(V_{PN}) \tag{8}$$

$$f(\gamma) = \left[1 + k_2 k_{12} \sin\left(\gamma \frac{\pi/2}{\gamma_0}\right)\right]^2 \tag{9}$$

$$R_T = 1 - k_4 V_P \sin \gamma$$

$$f(V_{PN}) = k_3 (V_P \sin \gamma)^4$$

$$k_2 = \begin{cases} 1.0 \text{ if } \gamma \leq 2\gamma_0 \\ 0.0 \text{ if } \gamma > 2\gamma_0 \end{cases}$$

Here, $\gamma_0$ is the angle of maximum erosion, $k_1$ to $k_4$, $k_{12,}$ and $\gamma_0$ are model constants and depend on the particle/wall material combination.

## 2.2. Simulation Geometry Model
### 2.2.1. Geometric Model Set Up

Figure 1 shows the flow diagram in the runner of the hydraulic turbine. The clearance flow between the runner and the fixed parts is complex, which can easily cause sediment abrasion.

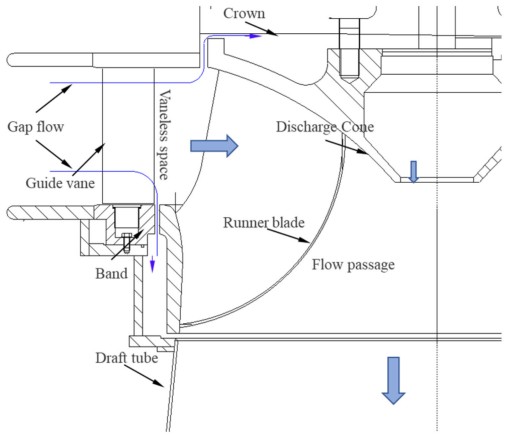

**Figure 1.** Flow diagram in runner.

A three-dimensional calculation model of the whole flow channel including volute, fixed guide vane, guide vane, runner, upper crown cavity, lower ring cavity, draft tube and outlet reservoir is established, as shown in the Figure 2.

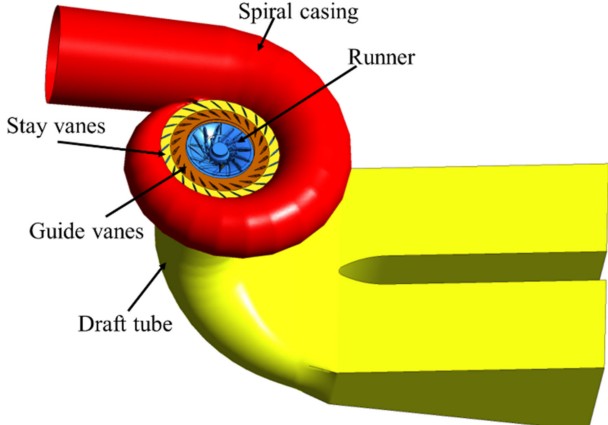

**Figure 2.** Model of numerical simulation.

The computational geometric model is meshed; a structured mesh is adopted for the gap and a hybrid mesh is adopted for other parts, as shown in Figure 3. The efficiency of hydraulic turbine is selected as the criterion for grid independence verification. Figure 4 clearly shows the relationship between turbine efficiency and grid number. When the number of grids increases from 11.2 million to 12.3 million, the absolute increment of turbine efficiency at the optimal operating point is less than 0.01%, so 11.2 million grids are used for the numerical calculation.

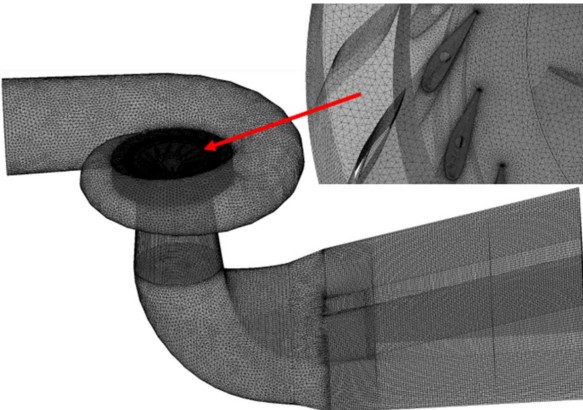

**Figure 3.** Clearance grids.

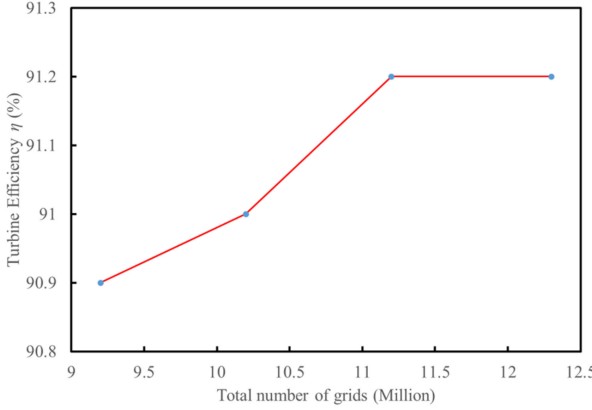

**Figure 4.** Efficiency of turbine with different grid numbers.

### 2.2.2. Parameter Setting in Calculation Model

(1)　Boundary conditions

In this paper, the Ansys CFX software is used to simulate the flow of sand in the full channel of hydraulic turbines with different guide vane openings. The inlet boundary was set to the total pressure corresponding to the water head of the upstream, and the outlet boundary was adopted the static pressure condition related to the water level of downstream. The wall of the unit adopted the no-slip boundary. The interface between the runner and stationary parts adopts the dynamic–static interface.

(2)　Calculation parameters

In the calculation scheme, the particle concentration is $10\ \mathrm{kg/m^3}$, the guide vane openings are $15°$, $20°$, and $30°$; among these values, $20°$ of opening is the optimal opening (see Figure 5). The solution precision is set to $10^{-5}$. The interphase coupling between the fluid phase and the solid phase adopted the one-way coupling for the low particle concentration.

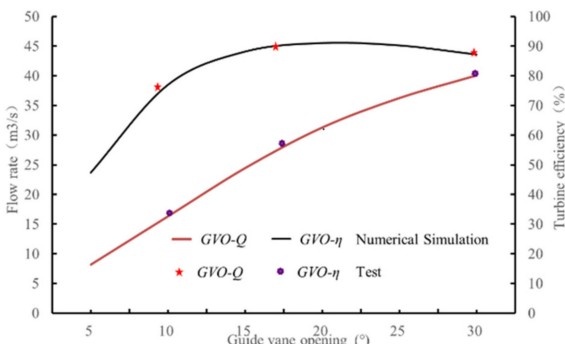

**Figure 5.** Reliability verification of external characteristics of hydraulic turbine.

*2.3. Reliability Verification of Calculation Model*

The reliability of the simulated results is verified by the field sediment erosion diagram of the runner of the hydraulic turbine unit and the field operation data of the unit. Figure 5 shows the calculated wear prediction and field wear comparison of the unit clearance and runner blades. The reliability of the calculation results is verified by the field wear diagram of the runner of the hydraulic turbine unit and the field operation data of the unit. The operation efficiency of the unit is obtained through steady calculation of the unit under different guide vane openings. Figure 6 shows the calculated wear prediction and field wear comparison of the unit clearance and runner blades. The clearance wear is consistent with the wear characteristics of water turbine on site. Figure 5 shows the operation curve of unit, and the tested data of unit operation is provided by Tagake Hydropower Station. The predicted results of operation characteristics and sediment erosion show that the numerical simulation method is reliable.

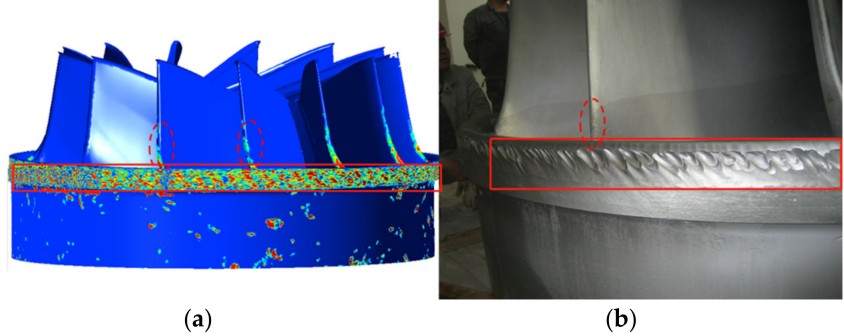

(**a**)  (**b**)

**Figure 6.** Sediment erosion at the clearance of the band. (**a**) CFD result; (**b**) physical wear picture.

## 3. Result Analysis

*3.1. Flow Pattern of Turbine under Different Openings*

Select the flow distribution on the central longitudinal section of the turbine to analyze the flow pattern in the turbine with different guide vane openings, as shown in Figure 7. When the guide vane opening is 15°, the flow pattern along the turbine is very poor, the flow velocity in the runner is low, and there are backflow and vortices in the draft tube. As shown in Figure 7a, these vortices and backflow increase the instability of the water flow and lead to the vibration of the unit. When the guide vane opening is 20°, the flow pattern in the turbine is obviously better than that in the turbine with the guide vane opening of 15°, as shown in Figure 7b. When the guide vane opening is 30°, the flow in the runner is very smooth, the flow velocity in the runner is the highest, and there is no backflow or vortices in the draft tube, as shown in Figure 7c. However, the increase in the flow velocity will also lead to more hydraulic loss and higher impact velocity of sand particles.

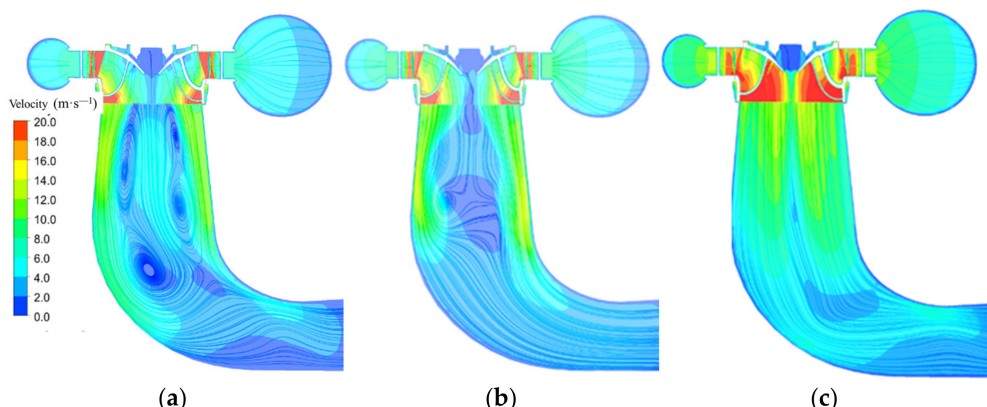

**Figure 7.** Overall flow pattern of internal section of hydraulic turbine with different opening. (**a**) 15°
(**b**) 20° (**c**) 30°.

Figures 8–10 shows the flow pattern in the guide vane and the runner on the horizontal section passing through the center of the runner. It can be seen from the figure that due to the small opening, the velocity triangle of the water flow at the inlet of the blade head completely deviates from the design velocity triangle of the hydraulic turbine, the velocity circulation of the guide vane outlet cannot meet the velocity circulation requirements of the runner blade, and the relative velocity angle of the water flow entering the runner is larger than the inlet angle of the runner blade, resulting in the water flow hitting the pressure surface of the runner blade at a large angle. The flow water loses the constraint effect of the blade, forms a blade passage vortex in the runner, increases the hydraulic loss of the runner blade, and reduces the hydraulic power generation efficiency of the turbine.

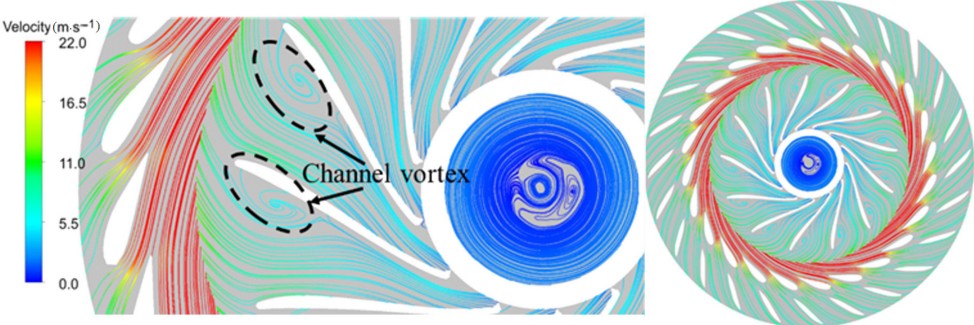

**Figure 8.** Flow pattern of vaneless space at 15 degrees of opening.

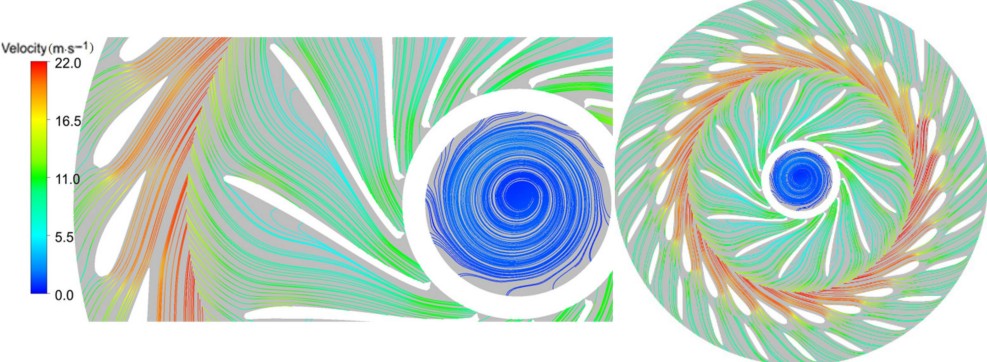

**Figure 9.** Flow pattern of vaneless space at 20° of opening.

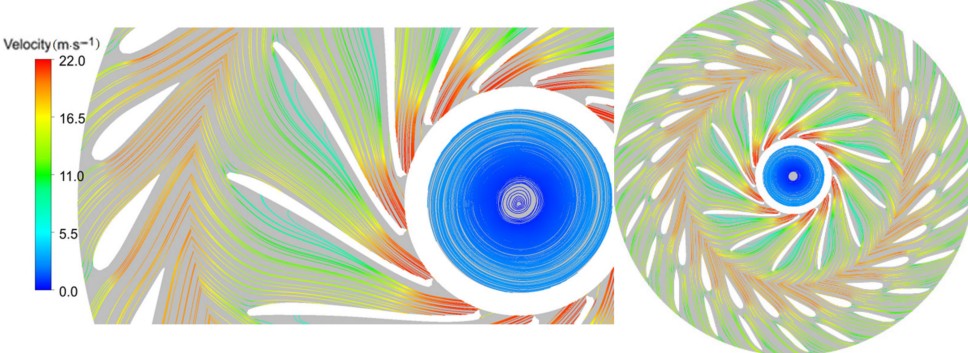

**Figure 10.** Flow pattern of vaneless space zone at 30° of opening.

### 3.2. Sediment Distribution at Different Guide Vane Openings

In order to further explore the flow characteristics and sediment distribution characteristics in the vaneless area, the particle distribution and solid volume distribution number distribution in the local vaneless area are selected for analysis. The horizontal section passing through the center of the runner is as shown in Figure 11.

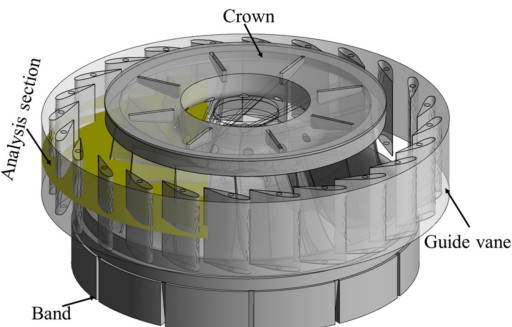

**Figure 11.** Schematic diagram of analysis section.

The sediment distribution in the guide vane and runner channel under different guide vane opening is shown in Figure 12. Figure 12 clearly shows that the sediment concentration in the guide vane channel and runner channel gradually decreases with the increase in the guide vane opening. When the opening of the guide vane is 15°, there is obvious sediment accumulation in the leafless area between the guide vane and the runner. With the increase in the opening of the guide vane, the sediment concentration in the channel of the guide vane gradually decreases, and the sediment is carried into the runner channel, which indicates that the increased opening of the guide vane will cause more wear and damage to the runner blade. However, with the increase in guide vane opening, the velocity of sediment particles increases, and the impact velocity on the guide vane and runner wall increases. Particles have more energy to destroy the wall structure of the runner and blade [21]. The smaller the opening, the higher the sediment concentration in the runner and guide vane channel. In the low-velocity region of the vortex return center, more particles are separated from the main flow into the vortex center of the channel and away from the runner wall [22,23], which is consistent with the flow change law in the runner and guide vane.

### 3.3. Sediment Erosion Distribution of Guide Vane and Runner Blade Wall

The sediment erosion distribution for the guide vane wall and runner blade wall under different guide vane openings is shown in Figure 13. There are different erosion rates on the guide vane and runner blade under different guide vane openings, and the wear is mainly distributed on the inlet head and outlet wall. When the guide vane opening of the runner is 20°, there is a little wear on the guide vane and the runner blade. Under this

opening, the wear of the guide vane only exists at the head of the guide vane, and there is no wear on the outlet surface of the guide vane. When the guide vane opening is 15° and 30°, the sediment erosion on the guide vane and runner blade is clearly serious, and the range and erosion rate of impact erosion and friction erosion are greatly increased, as shown in Figure 13a,b. This shows that the overall wear in the hydraulic turbine unit with the optimal opening is the smallest, the erosion rate increases under the small opening and the large opening, the wear of the guide vane and the runner inlet head is mainly impact wear, and the wear on the outlet wall is mainly friction wear, and the wear in the unit channel is the most serious under the large opening [24]. The long-term wear of the runner inlet and guide vane outlet will cause the loss of local structure in the bladeless area, increase the radial clearance of the runner, increase the clearance leakage, increase the vibration of the unit, and reduce the efficiency [25,26].

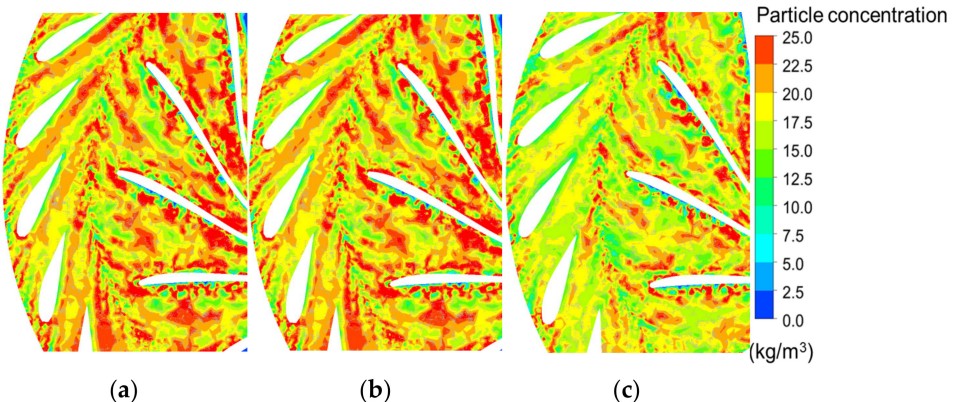

**Figure 12.** Volume fraction distribution of local solid phase in guide vane and runner passage: (**a**) 15°, (**b**) 20°, (**c**) 30°.

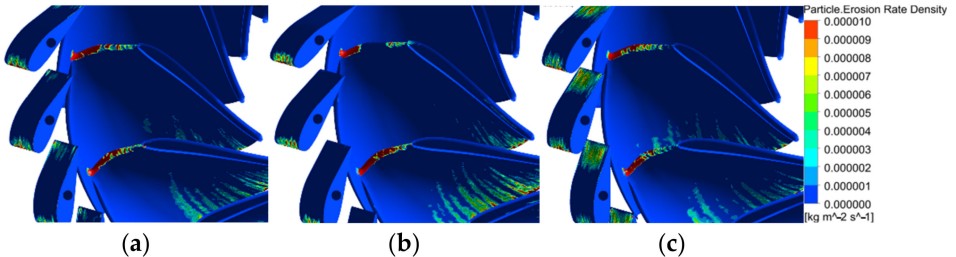

**Figure 13.** Sediment erosion distribution of guide blade and runner blade under different openings: (**a**) 15°, (**b**) 20°, (**c**) 30°.

## 4. Conclusions

This paper reveals the influence of radial clearance wear of a Francis pump turbine runner on the surrounding flow field characteristics under different guide vane openings, and obtains the following conclusions:

(1) The increase in guide vane opening has an important effect on the particle motion in the runner. With the increase in guide vane opening, the velocity of sediment particles increases, and the impact velocity on the guide vane and runner wall increases. The particles have more energy to destroy the wall structure of the runner and blade. The smaller the opening, the higher the sediment concentration in the runner and guide vane channel. In the low-velocity region of the vortex return center, more particles are separated from the main flow into the vortex center of the channel and away from the runner wall. This is consistent with the flow change law in the runner and guide vane.

(2) The opening of the guide vane affects the flow in the hydraulic turbine channel. When the opening of the guide vane is small, the velocity triangle of the water flow at the

inlet of the blade head completely deviates from the design velocity triangle of the hydraulic turbine, the velocity loop of the water flow out of the guide vane cannot meet the velocity loop required by the runner blade, and the relative velocity angle of the water flow entering the runner is larger than the inlet angle of the runner blade, resulting in the water flow hitting the pressure surface of the runner blade at a large angle, and the flow phenomenon occurs at the head of the runner blade. The separated flow loses the restriction of the blade and forms a blade passage vortex in the runner.

(3) The overall sediment erosion in the hydraulic turbine unit with the optimal opening is the smallest, and the erosion rate increases under a small opening and a large opening. The wear of the guide vane and the runner inlet head is mainly impact wear, and the sediment erosion on the outlet wall is mainly frictional wear. The sediment erosion in the unit channel is most serious under a large opening. The long-term wear of the runner inlet and guide vane outlet will cause the loss of local structure, increase the radial clearance of the runner, increase the clearance leakage, increase the vibration of the unit, and reduce the efficiency.

**Author Contributions:** Data curation, X.S.; software, validation, Z.J.; formal analysis, investigation, A.Z.; resources, F.S.; writing—original draft preparation, X.S.; writing—review and editing, Z.W. All authors have read and agreed to the published version of the manuscript.

**Funding:** This work was supported by a research project on sediment abrasion mechanism of turbine and R&D of anti-abrasion runners in Tagake Hydropower Station [XHTS-A-WZ-2022-005], National Natural Science Foundation of China (51876099).

**Informed Consent Statement:** Not applicable.

**Data Availability Statement:** Not applicable.

**Conflicts of Interest:** The authors declare no conflict of interest.

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
