# Peer review of "Prediction for the Influence of Guide Vane Opening on the Radial Clearance Sediment Erosion of Runner in a Francis Turbine"

_water, doi:10.3390/w14203268_

Round 1

Reviewer 1 Report

Very good paper for the international community

Author Response

Dear Editors and Reviewers:

Thank you for your letter and for the reviewers’ comments concerning our manuscript entitled “Prediction for the influence of guide vane opening on the radial clearance sediment erosion of runner in a Francis turbine” (ID: Water-1947217). Those comments are all valuable and very helpful for revising and improving our paper, as well as the important guiding significance to our researches. We have studied comments carefully and have made correction which we hope meet with approval. Revised portion are marked in the paper. 

Reviewer 2 Report

Some minor text corrections-

line 110                  Strain rate

 line 139                 wared?  correct to your intended word meaning

line 206                   out   correct to outlet

line 250                   a view of wear    correct  from  your  'a few of wear' 

More Introduction discussion on particle size distributions from different Chinese rivers and how this affects blade wear would be of interest if available. Also some comments on mean wear depth progress of main turbine components as a function of turbine operation time would add depth to the discussion.  As wear progresses and blade roughness increases, some comments about loss of energy transfer efficiency would be interesting as this would be of interest to predict maintenance blade replacement time. These general comments, if addressed, would add more interest to engineers interested in turbines and what affects their lifetime as governed by other factors that affect wear rates beyond turbine geometry changes.

Author Response

(The authors gave the same response as above.)

Reviewer 3 Report

(1)     The introduction part lacks the summative language, especially the last paragraph of this part lacks the summary of the work done in the paper.

(2)     Please note the formatting and grammar errors in the article. For example, the unit in line 84. Please check whether similar errors exist in the article.

(3)     The author's research topic is very interesting, however, the paper lacks the relevant design parameters of the geometric model. Simple and necessary parameter examples are appropriate and essential.

(4)     In the chapter "2.2.2 Parameter setting in calculation model", among the three guide vane openings of 15°, 20° and 30°, the author has designated 20 ° as the optimal opening, but the author has not done any verification work in this regard before, please explain. In addition, the following research on three guide vane openings cannot be convincing.

(5)     In figures 7.8.9 and 10, if the attributes represented by Contour Legend are the same. Suggested by one legend only. The form shown in Figure 12 is desirable.

(6)     Contour Legend representation in Figure 13 is not clear, please make the same modification as suggestion 5.

(7)     The Euler-Lagrange method is used in this paper, and the author does not introduce the calculation setup in detail. Please provide supplementary and explanation.

Author Response

(The authors gave the same response as above.)

Reviewer 4 Report

The authors presented a numerical simulation to study the effects of guide vane opening ratio on the sediment erosion in a hydraulic turbine. Generally speaking, the manuscript was well organized and mostly well written. Nevertheless, a few shortcomings should be pointed out and it would be better if the authors could well address this minor revision before recommendation of acceptance.

Specifically:

1.      A detailed description of the particle motions model used in this study should be supplemented. Eq. (5) is a general introduction, lacking of detailed information focusing on this study, including which items were considered, how they were estimated, which were neglected and why was that.

2.      What is the definition of Turbine efficiency, and how it was calculated in numerical simulation.

3.      It would be better to have a schematic figure to clearly mark out what is the guide vane opening.

4.      Figure 6. The legend is partially wrong.

5.      Figure 13. The color bar and font are too small to recognize.

6.      The current conclusion is all qualitative description, indicating 20 is the optimal opening? Quantitative conclusion is recommended to be presented directly. If the authors concern that this result is dependent to many other parameters like the structural design of the turbine, is it possible to conduct a non-dimensional analysis, so that to improve the value of this work by expanding the applicable range?

7.      Some minor errors should be carefully checked and avoided:

Line 84: “m3/s”

Eq. (1): the font style is not unified.

Line 106: “k” should be italic.

Line 107: “equation 2”, should it be referred as “Eq. (2)”?

Line 109: “Where”, should it be “where”?

Line 132: “VPN” should be italic?

……

In total, I would like to recommend the authors process a minor revision before consideration of acceptance.

Author Response

(The authors gave the same response as above.)
